# Polymorphism in Toll-Like Receptors and *Helicobacter Pylori* Motility in Autoimmune Atrophic Gastritis and Gastric Cancer

**DOI:** 10.3390/cancers11050648

**Published:** 2019-05-10

**Authors:** Valli De Re, Ombretta Repetto, Mariangela De Zorzi, Mariateresa Casarotto, Massimo Tedeschi, Paolo Giuffrida, Marco Vincenzo Lenti, Raffaella Magris, Gianmaria Miolo, Cinzia Mazzon, Giorgio Zanette, Lara Alessandrini, Vincenzo Canzonieri, Laura Caggiari, Stefania Zanussi, Agostino Steffan, Antonio Di Sabatino, Renato Cannizzaro

**Affiliations:** 1Immunopatologia e Biomarcatori Oncologici/Bio-proteomics facility, Centro di Riferimento Oncologico di Aviano (CRO), IRCCS, 33081 Aviano, Italy; orepetto@cro.it (O.R.); mdezorzi@cro.it (M.D.Z.); mtcasarotto@cro.it (M.C.); massimo.tedeschi@cro.it (M.T.); lcaggiari@cro.it (L.C.); szanussi@cro.it (S.Z.); asteffan@cro.it (A.S.); 2First Department of Internal Medicine, San Matteo Hospital Foundation, University of Pavia, 27100 Pavia, Italy; paolo.giuffrida01@universitadipavia.it (P.G.); marco.lenti@unipv.it (M.V.L.); a.disabatino@smatteo.pv.it (A.D.S.); 3Gastroenterologia Oncologica Sperimentale, Centro di Riferimento Oncologico di Aviano (CRO), IRCCS, 33081 Aviano, Italy; raffaella.magris@cro.it (R.M.); rcannizzaro@cro.it (R.C.); 4Preventive Medical Oncology, Centro di Riferimento Oncologico di Aviano (CRO), IRCCS, 33081 Aviano, Italy; gmiolo@cro.it; 5SSD Endocrinologia e malattie del metabolismo, Azienda per l’Assistenza Sanitaria 5 Friuli Occidentale, 33170 Pordenone, Italy; cinzia.mazzon@aas5.sanita.fvg.it (C.M.); giorgio.zanette@aas5.sanita.fvg.it (G.Z.); 6Pathology Dept CRO Aviano, IRCCS, National Cancer Institute, 33081 Aviano, Italy; lara.alessandrini@aopd.veneto.it (L.A.); vcanzonieri@cro.it (V.C.); 7Department of Medicine (DIMED), Pathology, Azienda Ospedaliera di Padova, 35121 Padova, Italy; 8Department of Medical, Surgical and Health Sciences, University of Trieste, 34127 Trieste, Italy

**Keywords:** autoimmune gastritis, flagellin A, *Helicobacter pylori*, toll-like receptor 5 (TLR5), toll-like receptor 9 (TLR9)

## Abstract

Autoimmune atrophic gastritis (AAG) is associated with an increased risk of certain types of gastric cancer (GC). *Helicobacter pylori* (*H. pylori*) infection may have a role in the induction and/or maintenance of AAG and GC. Toll-like receptors (TLR) are essential for *H. pylori* recognition and subsequent innate and adaptive immunity responses. This study therefore aimed to characterize TLR polymorphisms, and features of bacterial flagellin A in samples from patients with AAG (*n* = 67), GC (*n* = 114) and healthy donors (HD; *n* = 97). TLR5 rs5744174 C/C genotype was associated with GC, lower IgG anti *H. pylori* response and a higher *H. pylori* flagellin A abundance and motility. In a subset of patients with AAG, *H. pylori* strains showed a reduction of the flagellin A abundance and a moderate motility compared with strains from GC patients, a prerequisite for active colonization of the deeper layers of the mucosa, host immune response and inflammation. TLR9 rs5743836 T allele showed an association with serum gastrin G17. In conclusion, our study suggests that alterations of flaA protein, moderate motility in *H. pylori* and two polymorphisms in TLR5 and TLR9 may favor the onset of AAG and GC, at least in a subset of patients. These findings corroborate the function of pathogen–host cell interactions and responses, likely influencing the pathogenetic process.

## 1. Introduction

Autoimmune atrophic gastritis (AAG) is a relatively common form of gastritis [1] characterized by an immune response directed toward self-parietal cells [2]. AAG induces hypo-/a-chlorhydria leading to vitamin B12 deficiency and pernicious anemia over time. In the late stages of the disease, an important reduction of the oxyntic glands accompanied by a reduction of the inflammatory reaction and the development of pseudo/hyperplastic polyps and/or intestinal metaplasia (IM) are present. Both the polyps and IM are associated with the development of gastric cancer (GC). AAG patients are known to be at increased risk of GC compared to the general population, although cohort studies of patients with AAG are limited. A recent study, although presenting limitations, reports an annual incidence of GC of about 1.4% in Americans with a biopsy-proven AAG on endoscopic gastric biopsy between January 2010 and November 2015; an incidence that was higher when compared with the estimated 0.07% annual-incidence of GC in the United States [3].

Traditionally, AAG has been considered as a disease affecting predominantly elderly female individuals of Northern European descent, but subsequent studies demonstrated that people of any age, gender, and ethnicity could be affected [3]. Similarly to other autoimmune diseases, AAG is more prevalent in women (F:M ratio 3:1) with an increasing incidence over the last years. To date, AAG pathogenesis remains poorly understood, although some pathways related to *Helicobacter pylori* (*H. pylori*)-driven gastritis, [4] and antigen presentation associated with the loss of tolerance have been reported. Recently we demonstrated a significant decrease in flagellin A (FlaA) protein expression in *H. pylori* strains isolated from patients with AAG compared to those isolated from patients with GC [5,6], the consequence of that was deeper in the present study.

Multifactorial etiology plays a role in GC development, although the genetic and immunological components still need to be fully explained. In most cases, especially of intestinal GC type, GC is associated with chronic *H. pylori* infection and represents the final stage of a multistep carcinogenic process including non-atrophic gastritis, chronic atrophic gastritis, IM, dysplasia and GC. Recently, although incidence trend studies suggest an evolution in GC carcinogenesis, with *H. pylori* infection no longer being the sole etiological driver [7]. Hence, GC incidence rate may increase in the next few years, mitigating the previously described predilection of male sex.

Although AAG and GC share a presumed immune-mediated pathogenesis as well as a possible association with *H. pylori* infection mechanism differences that distinguish or relate these two disorders remain to be elucidated. Starting from this premise, our study aimed to compare selected genetic toll-like receptor (TLR) polymorphisms functionally related with microbioma and host immune response and bacterial flagellin A (FlaA) characteristics in patients with AAG, GC and healthy donors (HD).

## 2. Results

### 2.1. Design of the Study and Patient’s Characteristics

Based on the diagnosis, participants were divided into three sets: GC, AAG and HD. The main steps of the study were: (1) the characterization of selected TLR polymorphisms associated with GC, AAG and HD, (2) the association of discriminating TLR5 and TLR9 polymorphisms with serum level of pro-inflammatory *H. pylori*—induced Il-8 and IL-18 cytokines, with the serum gastrin G-17 and pepsinogen (PG)I for diagnosis of premalignant gastric lesions and risk factors for GC development (3) the characterization of *H. pylori* isolated from AAG and GC regarding the flagellin flaA abundance and sequence information, the bacteria motility rate and the presence of the virulent CagA gene, (4) the potential association of data obtained with the TLR5 and TLR9 polymorphisms. Study design is schematically shown in Figure 1.

Main characteristics of patients are reported in Table 1. As expected, in the AAG group there was a clear gender difference in prevalence, whereby female individuals (77.6%) were generally more frequently affected than males. Conversely, male sex was more prevalent in the GC group (62.3%). GC was mainly of a distal location (63.3%), of the intestinal type (50.0%) and at late stages (73.3% of T3–T4, 70.0% lymphnode-positive, 17.5% M1). Patients affected by AAG also had pernicious anemia in 29.4% of cases. Familial clustering for GC was observed in 12.7% of cases.

### 2.2. Individual TLR-Polymorphisms Showed an Association of TLR5 rs5744174 Genotype-C/T and TLR9 rs5743836 Allele-T in AAG Patients Compared to Healthy Donors

Allelic and genotype frequencies were estimated for each selected TLR based on literature for their potential influence on *H. pylori infection* and/or GC susceptibility (Table 2).

The distribution of genotypes for all the TLR polymorphism was consistent with Hardy–Weinberg equilibrium. TLR8 gene is located on the X-chromosome and therefore its frequency is higher in AAG since female-prevalence is higher in AAG (77.6%) than in GC cases (38.1%. *p* < 0.0001). The distribution of genotype and allele frequencies in AAG, HD and GC of the TLR polymorphisms are shown in Appendix A. Allele and genotype frequencies observed in HD are consistent with literature (Caucasians HapMap information-Phase II CEU- for TLR1, TLR2, TLR4, TLR5, TLR8, TLR9, TLR2-196 to -174 del). No significant differences were found in the distribution of allele and genotype frequencies of all SNPs between patients with GC and HD (regression analysis). Conversely, the most significant polymorphism based on Akaike’s information criterion (AIC) using genetic inheritance model of association (SNPstats tool), was an over dominant model for the TLR5 rs5744174 polymorphism (T/T + C/C) in HD (61%) compared to AAG (45%) (*p* = 0.021 corrected by sex and age), and a higher frequency of the recessive C/C genotype in GC (20%) compared to AAG (10%, *p* = 0.03 adjusted by sex and age, Appendix A). In addition, AAG demonstrated a decrease in TLR9 rs5743836 allele C frequency (9%) compared to GC (17%) (*p* = 0.04, Appendix A).

### 2.3. Targeted H. pylori Flagellin Protein Identification and Quantification by LC-MS/MS Spectrometry

In a previous study [6], by two-dimensional fluorescence difference gel electrophoresis (DIGE) we identified a reduction in *H. pylori* flagellin A (flaA) protein abundance in the *H. pylori* isolates from AAG patients compared to those of GC. In the present study, we added *H. pylori* isolates from seven further AAG patients and repeated the analysis. The abundance of flaA of *H. pylori* isolates from AAG samples confirmed the selective reduction of the flaA in *H. pylori* isolates of AAG (*n* = 11, median 0.37) compared to those of GC (*n* = 18, median 0.90) (Figure 2A). Among these *H. pylori* isolates we selected two samples from AAG either due to their high (AAG-HP sample 961, Log St Abund. = 0.24), or low (AAG-HP sample 959, Log St Abund. = −1) flagellin A content and one sample of *H. pylori* isolate from a GC with a low flagellin A content (GC-HP sample 992, Log St Abund. = −0.79). Protein fluorescent one-dimensional electrophoresis (1DE) sample load was shown in Figure 2B. After immunoblotting (the whole blot is displayed in Appendix A), a band at ~54 kDa in chemiluminescence, corresponding to the weight of flagellin (flaA: UniProtKB entry P0A0S1, 53284 Da; flaB: UniProtKB entry Q07911, 53882 Da), was visible in all the three lanes (Figure 2C). The apparent discrepancy obtained between the immunoblotting data for flaA abundance band and the DIGE analysis may come from the affinity and specificity of the antibody which recognizes all types of flagellins and the ability of 2D-DIGE to discriminate among post-translational forms of the proteins. 

Identification of the protein components present in the band of about 54 kDa was performed by using liquid cromatography (LC) mass (MS)/MS spectrometry and extraction of peptides present in the 1DE gel in a range of ~50–60 kDa. Flagellin A peptides were identified in the sample 959 (gel portions nr 2), 961 (gel portions nr 3 and 4), and 992 (gel portions nr 5 and 6); in addition, peptides of flagellin type B were also detected in sample 961 (Figure 2D and Appendix A). Different flaA entries were found after database searches with Mascot software: the SwissProt 2::FLAA_HELPJ was found in all the samples, while the NCBInr 1::gi|261839570, 1::gi|317180372 and 1::gi|188527549 were found in the only sample 961, characterized by higher flaA (Appendix A).

### 2.4. DNA Sequencing of H. pylori Flagellin A

The sequence alignment of *H. pylori* flagellin FlaA of samples 961, 959 and 992, compared to a reference *H. pylori* FlaA gene; sample FlaA; *WP_000885488.1* demonstrated a high degree of similarity in their nucleotide sequences (97% overall identity) (Figure 3A). The alignment emphasized the FlaA conservation and compared to the reference flaA showed three mutations (C296R; G315S; Q502H), which are present in all three samples, and a fourth mutation (A178T) present in sample AAG 961 only. All the mutations were external to the D1 domain known for the TLR5 binding site (indicated as interface for TLR5 in the Figure 3A,B). However, we cannot exclude that the Q502H mutation, present in the C-terminal D0 region, could have an effect on the stabilization of a flagellin-TLR5 dimeric signaling complex [15] or to be sensed by the intracellular NAIP5/NLCR4 inflammasome receptor [16]. Moreover, in two rodent models of gastritis, the C296R mutation had been described to affect structure and function of the protein FlaA [17]. The sample 961 showed the A179T, in the vicinity of the known glycosylated 181T residue [18]. In vitro *H. pylori* FlaA protein had been reported to lack the TLR5 activation [19].

### 2.5. Motility Assays of H. pylori Isolates

*H. pylori* motility was investigated by assaying the ability of the *H. pylori* isolates from AAG and GC patients to spread on soft agar plates (Figure 4A). The area of spreading of *H. pylori* isolates from 11 AAG (hollow circle) and from 8 GC patients (filled squares) (the *Y* axis, Figure 4B) were plotted with the FlaA abundance obtained using the respective *H. pylori* isolate (The *X* axis, Figure 4B). Linear regression formula (*Y* = 0.12 + 0.42*x*) reflected populations of *H. pylori* from which abundance of the *FlaA* protein directly correlate with the area of *H. pylori spread*. Patients carrying the TLR5 rs5744174 C/C genotype showed an increased area of spreading, compared with T/T homozigotes (ANCOVA test, covariate FlaA abundance, *p* = 0.039, Figure 4C).

### 2.6. Identification of the CagA Virulent Gene in H. pylori Isolates

CagA gene was identified in all the *H. pylori* isolates of 18 GC patients (18/18, 100%) but in 4 of AAG patients (4/7, 57%). *H. pylori* isolates of AAG patients showed a CagA-positive gene in 3/4 (75%) carriers with the TLR5 C/T-TLR9 allele-T haplotype, and in 1/3 case (33%) of carriers having the TLR5 T/T-TLR9 allele-C haplotype.

### 2.7. Patients Having TLR5 rs5744174 T Allele Show a Tendency to Have a Higher H. pylori Antibody Titer and H. pylori Strains with a Lower Flagellin an Abundance

To further investigate the association of TLR5 rs5744174 polymorphism with *H. pylori* infection, we tested the impact of TLR5 genotype stratification on 1) IgG anti-*H. pylori* titer and 2) flagellin A abundance. Of the total of 104 cases tested for *H. pylori* infection 50 (34 GC and 16 AAG) were *H. pylori*-positive. *H. pylori*-positive cases in GC carrying the TLR5 T/T genotype (53.6%) were higher than in AAG (35.3%, Figure 5) and both in the respective groups showed a tendency to have an increased IgG anti-*H. pylori* titer as compared with carriers of C/C and C/T genotype, although the result did not attain a statistical significance (Figure 5)

In addition, the result of ANOVA test showed a significant variation between abundance of the spot, corresponding to the flaA protein in DIGE and GC groups stratified by different TLR5 genotypes (Figure 5B). In particular patients having the TLR5 T/T genotype showed a lower mean of the flagellin A abundance (−0.1771, SD ± 0.41) compared to those having the C/C (0.335 SD ± 0.017, *p* = 0.04) and the C/T (0.260 SD ± 0.08, *p* = 0.02) genotype. The number of AAG patients with FlaA abundance data available were too low to perform the analysis. 

### 2.8. Evaluation of the Serum Level of Inflammation-Related IL-8 and IL-18 Cytokines

TLR activation results in the release of various chemokines and cytokines. We focused on the effect of TLR5 and TLR9 polymorphisms on the pro-inflammatory cytokines interleukin (IL)-8 and IL-18, sharing a common relation to *H. pylori* infection. TLR5 polymorphism showed no significant relation with neither IL-8 nor with IL-18 production in GC patients. Conversely, these results showed higher amounts of secreted serum IL-8 in patients having the TLR9 rs5743836 T/T genotype (4.21 ± 3.18 pg/mL, mean ± SD) compared to those having the C/T or C/C genotype (1.63 ± 1.58 pg/mL, mean ± SD) (*p* = 0.015; Figure 6A). A similar effect, which, however, did not reach a statistically significant difference, was observed for the IL-18 (Figure 6B).

### 2.9. Association of the Serum Level of Atrophic Markers Gastrin G17 and PGI with TLR9 rs5743836 T Allele

Elevated gastrin G17 titer is a diagnostic marker of AAG and in association with low PGI level (<48 pg/mL), well correlate with advanced stage of metaplasia in AAG, a condition considered to be at risk for GC development [20]. We focused if TLR5 or TLR9 polymorphisms were associated with higher level of the released serum AAG marker gastrin G17 and/or a lower level of PGI. Results showed higher mean of secreted gastrin G17 level in AAG patients (Figure 6C), but not in GC (Figure 6E), having the TLR9 rs5743836 T/T (mean 376.52 SD ± 339; C/T or C/C genotype mean 137.52, SD ± 124; *p* = 0.02).In both the AAG and GC patients, the difference between TLR9 genotype and PGI titer was slightly higher in the patients with the T/T genotype (mean 34.75 SD ± 52 in AAG, Figure 6D; 161.53 SD ± 161 in GC, Figure 6F, respectively) compared to those with the T/C or C/C genotype (18.42 SD ± 14.56 in AAG; 153.74 SD ± 113 in GC), but in these last cases the difference did not reach a statistical significance.

## 3. Discussion

Alterations in the immune response of patients with AAG may predispose to GC, and a link between AAG and GC has been reported in many studies. AAG is the outcome of a pathological CD4 T cell-mediated autoimmune response directed against the gastric H+/K+-ATPase and several epidemiological data point to a relation between immunogenetic background of the host and AAG development. A role for *H. pylori* as an antigen trigger for the genesis of AAG trough a molecular mimicry between *H. pylori* antigens and gastric H/K-ATPase was evoked, although evidence to support that this pathogen is needed for disease development still require more extensive investigation [21,22].

In the present study, compared with HD, we found that the TLR5 rs5744174 C-allele and the reduction of *H. pylori* motility were associated with AAG. Furthermore, results showed a TLR5 discrepancy between AAG and GC with a predominance of TLR5 rs5744174 in heterozygosis (C/T) in AAG, and in homozygosis (C/C) in GC suggesting a possible role of TLR5 C-allele in increasing the risk for gastric diseases. In addition, the lower frequency of TLR9 C-allele in AAG, largely of female sex, compared to GC, suggested an association with a protective sexual dimorphism related to TLR9 in AAG.

The TLR5 receptor specifically binds bacterial flagellin, the principal component of flagella, a bacterial appendage necessary for its locomotion. Activation of TLR5, located on the cell membrane surface, by flagellin results in activation of the NF-kB pathway, which is linked to cancer, inflammatory and autoimmune diseases. Due to the high importance of TLR5 in induction of both innate and adaptive immunity flagellin is now used as an adjuvant of different vaccine preparations and an anti-cancer agent in several clinical trials, as well as a potential agent to reduce radiation toxicity. However, besides having beneficial actions, it is also increasingly clear that a paradoxical effect of TLR5 expression may be present in different autoimmune and inflammatory diseases (e.g., acute lung inflammation, inflammatory bowel disease, liver injury) [23,24] as well as in specific cancers, as in the case of GC [25,26].

On the contrary to other flagellated bacteria, *H. pylori* flagellin, like those of *H. pylori* strains isolated from the biopsies of our patients (Figure 3), presented a peculiar binding interface for TLR5 (D1 and D0 domains, Figure 3) did not consent the traditional TLR5 activation [19], but nonetheless may produce an inflammatory complex [27]. Indeed, it was demonstrate that the peculiar D0 domain of *H. pylori* flagellin, interacting with the C-terminal region of the TLR5 including the TLR5 rs5744174 polymorphism and playing a role in TLR5 dimerization [15], may be recognized by both the TLR5 and the intracellular Nod-like receptors, NLRC4, an element of inflammasome complex. However, the *H. pylori* flagellin- TLR5/NLCR4 recognition failed to elicit the caspase-1 activation necessary to processes pro-IL-18 into IL-18-mature form of secretion and to initiate pyroptosis cell-mediated death [27,28]. In spite of that, both TLR5 and NLRC4 receptors contribute to flagellin-induced antibody production (adaptive immunity) [29], since TLR5 may present peptides to flagellin-specific CD4+ T-cells also in the absence of conventional TLR signaling [30]. In our series, the different TLR5 genotypes we found showed a similar serum IL-8 and IL-18 cytokine level among patients, while the TLR5 C-allele was associated to a reduction in the *H. pylori*-induced antibody production (Figure 5A). Thus, these results indicate that in our patients TLR5 in the presence of *H. pylori* flagellin should produce an inflammasome complex that however did not culminate in the release of IL-18 involved in resistance to infectious pathogens, and that moreover the presence of TLR5 C-allele, particularly when homozygous, might reduce the host adaptive immune response (anti *H. pylori* antibody), thus favoring *H. pylori* persistence and tissue damage-associated inflammasome activation. In accordance with this, *H. pylori* flagellin by influencing the inflammatory milieu trough TLR5 might have a role in gastric cancerogenesis, a hypothesis supported by the demonstration that inflammasome-derived exosomes from GC cells were able to directly activate NF-kB signaling pathway promoting GC [31] and as a consequence of the inflammation, the disruption of the epithelial polarity and integrity of the gastric cells should favor the insertion of *H. pylori* into the gastric epithelial cells [32], a phenomenon also associated with GC development.

Previously we found in *H. pylori* strains isolated from AAG patients a significant decrease in the flaA protein compared to strains isolated from both GC and from patients with duodenal ulcer [33]. Herein, we confirmed that *H. pylori* strains of AAG showed a reduction in the flaA abundance (Figure 2A). Furthermore, we also demonstrated a moderate reduction in motility of those *H pylori* strains showing a reduction in the FlaA abundance or with a mutation in the FlaA sequence (sample 661). Presence of TLR5 T-allele also correlate with the reduction of flaA abundance and *H. pylori* motility (Figure 4). Several studies have demonstrated that alteration in the flaA expression as well as its glycosilation modulated the motility of *H. pylori* [34,35]. Phenotypic and molecular characterization further demonstrated that *H. pylori* strains showing the highest motility exhibited a better capability of colonization and cancerogenesis (CagA phosphorylation and NF-kB activation) [35,36]. Indeed, *H. pylori* needs to invade and proliferate in the epithelial cells to induce GC development [37]. Thus, the reduction in motility, although moderate, showed in *H. pylori* strains isolated from patients with AAG, also more evident in patients carrying the TLR5 T-allele, should protect patients by reducing the efficiency of *H. pylori* colonization and the maintenance of infection in the gastric epithelium. However, modulation of the motility is a result of several factors, not only flaA expression, as at least in one case of *H. pylori* with a relative high abundance of flaA expression isolated from an AAG individual (sample 961) we found a reduction in the motility in the in vitro assay (Figure 4). In this last case, however, a mutation in the flagellin A sequence near a site of glycosilation was found. In addition, *H. pylori* chronic infection could be reduced by the hypo-achlorhydria that characterizes the gastric milieu in AAG patients [38]. *H. pylori* adhesion to gastric cells is essential to inject the bacterial CagA oncoprotein into the host cells [36]. We found the CagA gene in the *H. pylori* strains isolated from all the GC cases while only in 50% of *H. pylori* isolated from AAG patients (Figure 4) and that the CagA gene was present less frequently in *H. pylori* isolated from AAG carrying the suggested protective TLR5 rs5744174 genotype-T/T. Therefore, moderate motility and a low frequency of CagA gene in *H. pylori* strain isolated from patients with AAG and TLR5 T/T genotype, both allowed the distinction between AAG and GC, becoming in these way potentially protective factors towards GC development in these peculiar subset of AAG patients (Figure 4). The reason for flaA and motility reduction in *H. pylori* isolates from AAG patients remains to be elucidated since the regulatory network that modulates the expression of flagellar genes and bacterial motility is complex and not fully understood. In the intestinal tract, TLR-mediated flagellin antibody response was found to originate from antigen only when a tissue damage condition was present [39] and the formation of the flaA-anibody complex undergoing rapid degradation via Fc receptor-mediated pahogytosis [40]. Therefore, we hypothesized that the higher *H. pylori*-antibody level we found in patients having the TLR5 T/T genotype could in part contribute to the protective reduction of flagellin A expression in some patients. Alternatively, *H. pylori* strain-gene specificity due to the high variation of the *H. pylori* genome, may regulate the flagellin gene expression and most likely induce severe disease. Further studies should be performed to elucidate this point.

In addition, the interaction between flagellin from other bacteria than *H. pylori* and TLR5 might have a role in gastric diseases. In vitro and in vivo studies showed that homozygous TLR5 rs5744174 C/C carriage is associated with a reduced traditional TLR5 responsiveness to flagellin of overall gastric commensal bacteria [41]. Accordingly, T/T carriage was associated with higher secretion of several antimicrobial factors against most infections resulting in a better protective genotype, compared to the C/C genotype [42]. In line with this hypothesis, it has been speculated that TLR5 rs5744174 variant may be informative not only for *H. pylori* but also for commensal bacteria whose contribution to immunomodulation is only emerging, suggesting that survival of gastric microbioma via TLR5 might not only be dependent on the species specific stimulatory potential of flagellin but also on TLR5 genotype [41]. As a consequence the resultant positive selection of the rs5744174 C-polymorphism among the human evolutionary changes in individuals constantly exposed to high levels of pathogens has been suggested to be likely to reduce NF-κB activation, which perpetuated a pro-inflammatory status and tissue damage in the host [41]. The TLR5 rs5744174 C/C genotype was also associated with increased risk for some cancers, including GC [26] and a lower survival in colorectal cancer by influencing cytokine response to intestinal bacteria [26] However, NF-κB paradoxically may also be involved in autoimmune suppression, by promoting the development and the immunosuppressive function of regulatory T cells, which are also crucial for central and peripheral tolerance, although this factor alone was not enough [43]. Intriguingly, over the recent decades in Western societies, concurrently to the overall reduction of infectious diseases and the consolidation of the TLR5 C-variant in the human genome, epidemiological data provided evidence of a steady rise of autoimmune diseases [44].

In the present study, we also found a decrease in the TLR9 rs5743836 minor allele frequency (i.e., -1237 C-variant) in AAG, compared to GC, suggesting a protective role of the TLR9 C-variant towards AAG development. The rs5743836 variant-C, located at the promoter region of the TLR9 gene creates IL-6 and NF-kB responding elements, so conferring an increased TLR9 transcription activity compared to the T-variant in response to various stimulants, like induced NF-kB-mediated inflammatory factors. To our knowledge, this is the first study regarding the TLR9 rs5743836 polymorphism in AAG. A clear sex difference in prevalence for most autoimmune diseases, including AAG, whereby female individuals are generally more frequently affected than male [45]. Recently, in healthy individuals, serum testosterone levels was found to negatively correlate with the expression of TLR9 suggesting that this effect may have a protective role against autoimmune disease development [46]. TLR9 was shown over-expressed in hormone-regulated cancers (e.g., breast, ovary and prostate cancers) where it showed by contrast a relationship with a poor prognosis; in these cases the increasing expression of estrogen receptor (ERα) have been reported to lead to the decreased expression and function of TLR9 with a beneficial effect against tumor invasion [47].

TLR9 expression in B cells was shown important for autoantibody production and activation of autoreactive B cells in the periphery and treatment by using TLR9 inhibitors was demonstrated to improve the clinical outcome of some autoimmune disorders (e.g., rheumatoid arthritis, systemic lupus erythematosus) [48]. Regarding *H. pylori* infection, in literature TLR9 activation resulted in a pro-inflammatory response [49], but during the acute phase of *H. pylori* infection, TLR9 showed an opposite role by seeming to promote anti-inflammatory signaling in order to favor the establishment of a persistent infection and, thus, acting as a suppressor of *H. pylori*-induced gastritis [50,51]. TLR9 expression also differs in healthy individuals and in patients infected with the *H. pylori*; TLR9 was located in the apical compartment of the gastric epithelium in healthy individuals, while its expression was located in the basolateral compartment in individuals having a *H. pylori* infection [32]. Authors proposed that the inflammatory microenvironment resulting from chronic *H. pylori* infection contains more gastric cells without polarity that activate TLR9 to further promote the inflammatory cascades and then eventually the development of GC. Accordingly, TLR9 was found up-regulated in GC tissue and absent in epithelium with intestinal metaplasia and dysplasia [32] and TLR9 C-allele, inducing a higher TLR9 expression, was found associated with GC [14], as well as in our series, in high GC risk areas, like Colombia [52], in relatives of GC patients in West of Scotland [14]. Cag T4SS system was mandatory to actively transport *H. pylori* DNA into the host cells and for TLR9 engagement [52]. In our series CagA gene was present in all GC cases but only in 50% of AAG cases supporting the active role of CagA gene and TLR9 activation in particular in GC. Moreover, we found an association between TLR9 rs5743836 C-allele with a lower PGI level, indicative of advanced stage of atrophy in AAG [20], with an increased TLR9 expression (Figure 6) and in GC with an increase in the pro-inflammatory cytokines, IL-8 (Figure 6), which is known to be induced by *H. pylori* infection in GC [14]. Overall these data suggested that TLR9 rs5743836 C-allele was less frequent in AAG, particularly of female sex, since TLR9 express a sex hormone or receptor that may modulate the TLR9 expression and that expression/activation of the TLR9 could result in a detrimental effect in GC development. A recent study showing that (i) expression of TLR9 and TCR inducible costimulatory receptor (ICOS) ligand (ICOS-L) in plasmacytoid dendritic cells (pDCs) infiltrating the GC tumor showed a strong immunosuppressive function and (ii) that the percentage of these cells increased in patients with *H. pylori* infection, more in the late GC stages than in the early stages and in patients with a lower relapse-free survival [53], support our results. As TLR9 is located in the cytoplasmic compartment, it is believed that ligand(s) may gain access to the TLR9 by using a receptor-mediated delivery (e.g., *H. pylori* T4SS system, B-cell receptor immune complexes, KIR3DL receptor).

To conclude, Figure 7 presents abridged form of the principal characteristics observed in AAG and GC in our series. Overall data indicate that the reduction of motility and TLR5 rs5744174 polymorphism may result in a lower probability of *H. pylori* to reach and infect gastric epithelium cells, thus protecting in part AAG individuals to GC development. In addition, TLR9 rs5743836 minor C allele, might have a protective role against AAG development, perhaps by decreasing expression of the TLR9 as a consequence of sexual hormone release. Further studies are necessary to elucidate these last points in AAG and GC.

## 4. Materials and Methods

### 4.1. Patient Characteristics and Ethics Statement (IRB-14-2013)

Overall, 278 Italian individuals attending the Centro di Riferimento Oncologico, IRCCS Aviano and the Fondazione IRCCS Policlinico San Matteo (Pavia, Italy) undergoing upper gastrointestinal endoscopy with gastric biopsies were included. After histological examination, individuals were grouped as follows: confirmed GC diagnosis (*n* = 114) and AAG (*n* = 67). Healthy blood donors (HD) without either GC or AAG (*n* = 97) were included as negative control. HD were individuals with functional dyspepsia, normal gastric histology and a negative family history of GC (*n* = 44) or blood sample donation from healthy individuals collected by the CRO biobanking for DNA research (*n* = 53). Histopathologic diagnosis of GC and AAG was confirmed by experienced gastrointestinal pathologists. GC was classified according to the Lauren classification [54] and the disease stage was assessed according to the TNM criteria, 7th edition [55]. Diagnosis of AAG was based on both the updated Sydney-Houston criteria (stomach corpus and fundus atrophy with antrum sparing) and the anti-parietal cell antibody positivity (PCA) [2]. Overall clinical, pathological and laboratory features of patients with AAG and GC, and those of HD were reported in Table 1. GC were Caucasian patients: 71 men and 43 women with a mean age of 61.45 ± 1.04 years (range 19 to 85 years). AAG patients were: 15 men and 52 women with a mean age of 54.59 ± 1.79 years (range 31 to 70 years). The healthy controls were: 55 men and 42 women with a mean age of 42.03 ± 1.66 years (range 24 to 64 years). Among these, 197 (60 GC, 40 AAG and 97 HD) were screened for pepsinogen (*PGI* and *PGII*), gastrin G17 and *H. pylori* infection using the gastropanel kit (Biohit HealthCare, Milan, Italy) [20]. Significant value for positive IgG anti-*H. pylori* antibody titer is ≥30 U/mL. Parietal cell antibody (PCA) titer was determined in all patients through immunofluorescence (IF) technique. A titer greater of 1:40 was considered as significant. In case of PCA negativity detected through IF, but with histopathological lesions consistent with AAG, PCA were assessed again with an ELISA technique, which is more accurate [56,57]. Only PCA positive patients were included in this study. Pernicious anemia with vitamin B12 deficiency or folate deficiency were diagnosed according to laboratory tests (hemoglobin <13 g/dL in men, <12 g/dL in women, serum vitamin B12 < 240 pg/mL, serum folate < 3 ng/mL) and occurred in 70.6% of AAG cases. The study was approved by the Internal review board (CRO no. 14). Informed consent was obtained for all subjects.

### 4.2. Selection of TLRs Genetic Variants

Eight TLR polymorphisms were selected based on literature for their potential influence on *H. pylori* infection and/or GC susceptibility. TLR2 and TLR4 have a key role in recognition of various structural components of bacterial outer membrane [9,10,11]. TLR1 and TLR6, represent coreceptors of TLR2 and associated with risk for atrophic gastritis [8]. TLR4 polymorphic variants were linked to a decreased responsiveness to *H. pylori* lipopolysaccharide and a risk factor for precancerous lesions; although other studies indicated TLR2 rather than TLR4 as the dominant innate immune receptor for the recognition of gastrointestinal *Helicobacter* species. Murine models indicated a role for TLR2, TLR9, and TLR8 in the recognition of *H. pylori* by dendritic cells, which are recruited during inflammation and have the ability to traverse the epithelial tight junctions to sample luminal bacterial antigens for presentation. TLR5 recognizes bacterial flagellin and a still unknown bacterial protein [58]. TLR3, 7, 8 and 9 are expressed in the endosomal membrane and can recognize DNA and RNA of invading microorganisms [59]. Some TLR8 polymorphisms has been associated with exacerbated gastric inflammation and were found more frequently in Asian (85%) than in Caucasian patients (20–30%) in agreement with the higher frequency of *H. pylori* infection in Asian populations than in American or European. TLR9 recognizes the DNA of cancer-associated CagA+ type IV secretion system (T4SS) of *H. pylori* so interfering with the cellular persistence of the bacteria [52]. Table 2 listed the selected TLR1, TLR2, TLR4, TLR5, TLR8 and TLR9 genetic variants used in the present study.

### 4.3. Analysis of TLR Polymorphisms

Genomic DNA was extracted from whole blood by using the EZ1 Qiagen blood kit (Qiagen Inc., Valencia, CA, USA). Allele-specific polymerase chain reaction (PCR) was used to detect TLR2-196 to -174del and TLR4 polymorphisms [60]. The primer sequences and the sizes of PCR products are shown on Table 3. The TLR1 (rs4833095), TLR2 (rs3804099), TLR8 (rs3764880) and TLR9 (rs5743836) SNPs genotyping were performed by allelic discrimination using TaqMan SNP-genotyping Assays (Applied Biosystems/Thermo Fisher Scientific, Waltham, MA, USA). Genotyping was carried out on an ABI 7900HT Fast Real-Time PCR System (Applied Biosystems, Foster City, CA, USA) with standard conditions and according to the manufacturer’s protocol. We used a home-made High-Resolution Melting method (HRM) to discriminate the TLR5 rs5744174 genotype. PCR-HRM analysis was performed using a CFX96 Touch Real-Time PCR Detection System (Bio-Rad, Hercules CA, USA) in a final volume of 20 μL of a reaction mixture containing: 30 ng of genomic DNA template and 0.7 µM of each PCR primer. The HRM-PCR protocol included 1 cycle at 98 °C for 3 min, followed by 40 cycles at 95 °C for 10 s, 62 °C for 10 s, and 72 °C for 20 s. HRM curve acquisition (60 °C to 95 °C, with an increment of 0.1 °C/10 s) and analysis were performed on CFX Real-time PCR instrument (Bio-Rad CFX Manager software). Melting curves were normalized to relative fluorescence units (RFU) in a specified melt region (77.5 °C to 82.5 °C). Three pre-defined genotype samples and a negative control without DNA were used as controls.

### 4.4. Anti-Flagellin Antibody and Liquid MS/MS Spectrometry Validation

In a previous study comparative 2D-DIGE analysis of *H. pylori* strains isolated from patients with AAG (AAG-HP) and GC (GC-HP) revealed a significant decrease in expression of the flaA protein in *H. pylori* strain isolated from AAG patients [6]. In the present study we want to further validate this data by adding some samples (Figure 2A) and by using immunoblot and liquid mass spectrometry analyses. Analyses focused on two samples of AAG-HP characterized by a high (sample 961, Log St Abund. = 0.24), or a low (sample 959, Log St Abund. = −1) flagellin A content. A sample of GC-HP showing a low flaA content was used as reference control (sample 992, Log St Abund. = −0.79). Ten µg of proteins were fractionated on 12% Criterion TGX Stain-Free gels and, after gel image acquisition with the Chemidoc system (Bio-Rad), electrotransferred onto nitrocellulose membranes. Membranes were incubated with the monoclonal antibody anti-flagellin (1:12000; ABIN235293, Antibodies Online, Aachen, Germany). Antibody-bound proteins were detected by enhanced chemiluminescence using the Chemidoc system after incubation with ECL HRP-conjugated secondary antibodies (1:10,000 dilution, Santa-Cruz, Dallas, TX, USA) and reaction with ClarityTM Western ECL Substrate (Bio-Rad). The image of the gel acquired before its transfer was used as control for equal protein loading among samples.

Protein extracts from AAG- and GC-HP (50 µg per lane) were separated by 1DE, and images of blue-stained gel were acquired with the Chemidoc system. A total of two gel portions in the MW range between 50 and 60 kDa containing proteins cross-reacting with the flagellin antibody (Figure 2D, rectangle and numbered lanes) were excised, reduced by incubation with 10 mM dithiothreitol (1 h at 57 °C), and alkylated with 55 mM iodoacetamide (45 min at room temperature). Samples were further washed with NH4HCO3, dehydrated, trypsin digested and processed for LC-MS/MS analyses using a LTQ XL-Orbitrap ETD equipped with a HPLC NanoEasy-PROXEON (Thermo Fisher Scientific, Waltham, MA, USA). Database searches were done with the MASCOT search engine version 2.3 against SwissProt and NCBInr (Matrix Science, London, UK), and the presence of flagellin was searched among first 30 report hits.

### 4.5. Flagellin A Nucleotide Sequencing

The nucleotide sequence of flaA encoding a protein of 510 AA with a predicted molecular mass of 53.2 kDa. Genomic DNA for flaA sequencing was extracted from *H. pylori* strains isolated from AAG and GC patients as previously reported [6]. The following samples were chosen based on the high variance in the flaA expression: sample 961 from AAG, with both FlaA and FlaB expression, sample 959 from AAG, with only FlaA expression, and sample 992 from a GC showing lower abundance of FlaA when compared with other *H. pylori* strains isolated from GC patients (Figure 2). After alignment of 30 non-redundant *H. pylori* flaA gene sequences(https://www.ncbi.nlm.nih.gov/), a conserved region was chosen and the Primer3Plus software was used to design the oligonucleotide primers for amplification and sequencing (http://bioinfo.ut.ee/primer3-0.4.0/). Sequences of primers were reported in Appendix A. Template DNA for cycle sequencing was of 1521 bp and obtained through amplification for 30 cycles as follows: 94 °C for 30 s, 48 °C for 30 s, and 72 °C for 2 min with an initial denaturing step of 95 °C for 6 min. The PCR products were purified by the addition of 3 μL of Clean up Reagent (Abbott, Des Plaines, IL, USA), 37 °C for 15 min and 80 °C for 15 min. Cycle sequencing was performed using DyeDeoxy Terminator Cycle Sequencing kit 3.1 from Applied Biosystems. PCR products were purified using the Centrisep spin columns (Princeton Separations, Adelphia, NJ, USA) and run on an ABI Prism 3130XL automated DNA Sequencer (Applied Biosystems, Foster City, CA, USA). Raw sequences of DNA samples were analyzed and compared to a reference *FlaA* aminoacid sequence from *H. pylori* genome (NCBI WP_000885488) using multiple BLAST CLUSTALW alignment analysis.

### 4.6. Motility Assays of H. pylori Isolates

For the motility assay *H. pylori* samples isolated from 11 patients with AAG and used for the flaA abundance analysis were chosen and compared with other *H. pylori* strains isolated from eight GC patients. Briefly, 3 uL of *H. pylori* suspensions grown at the same exponential rate and at a 10^8^ colony-forming units (CFU)/mL were spotted onto soft agar plates containing 4.3% Brucella Broth (Difco BD, Saprks, MS, USA), 0.4% of Bacto agar (Difco) and 10% horse serum and incubated for seven days at 37 °C under microaerophilic conditions. The motility assay was conducted in three independent experiments and the mean diameter of the migration was then calculated and compared to the reference *J99 H. pylori* strain.

### 4.7. Identification of the CagA Gene

*H. pylori* genomic DNA was extracted from *H. pylori* isolates from 18 GC patients and from 7 AAG as previously reported [6] and tested for the presence of the *CagA* virulent gene. Primers specific for CagA gene [61] were used for the PCR amplification and products were then analyzed on 8% polyacrylamide gel electrophoresis.

### 4.8. IL-8 and IL-18 Cytokines Assay

A serum sample was collected from all participants. Cytokines were assayed using a Luminex based MAG multiplex assay (Luminex 200 Xponent, Austin, TX, USA). Two cytokines (IL-8 and IL-18) from a custom Human High Sensitivity panel (ThermoFisher Scientific) were used in GC. Serum samples were thawed at 4 °C and then centrifuged at 1400× *g* to remove any protein aggregates that may potentially obstruct the measurement. The supernatant was then transferred to a fresh tube and was diluted 1:2 in assay buffer. A 10-point standard curve with serial dilutions of 1:4 was generated using reconstituted stock standards supplied by the manufacturer; quality controls supplied by the manufacturer were also used to determine assay accuracy (Luminex 200 performance and verification and calibration kits). The data was generated using the Xponent software, which calculated average values against a 5-parameter logistic standard curve corrected by background readings. Plasma samples were assayed in duplicate.

### 4.9. Statistical Analysis

Continuous variables are presented as the mean ± SD and were analyzed by independent *t*-test and Anova test among variable groups. Genetic data was assessed using regression analysis (SNPStats software; http://bioinfo.iconcologia.net/SNPstats) in patients and controls, and adjusted for sex. For binary responses, the logistic regression analysis was performed to obtain genotype frequencies, odd-rations (OR) and 95% confidence intervals (CI). Multiple logistic regression “enter” model, include variables which meet a preset cutoff for significance (*p* < 0.10) were employed to obtain odds ratios (ORs), 95% confidence intervals (CIs) and *p*-values. Five inheritance models are fitted which correspond to different parameterizations of the genotypes: co-dominant, dominant, recessive, over-dominant, and additive. Most frequent genotypes are automatically selected as reference category. The selection of the best model for a specific polymorphism was calculated using the Akaike’s information criterion (AIC). Statistical significance was set at *p* < 0.05.

## 5. Conclusions

In conclusion, our study suggests that a moderate motility in *H. pylori* and two polymorphisms in TLR5 and TLR9, may represent critical determinants in AAG and in GC development, at least in a subset of patients. Therefore, understanding of the complex regulatory pathways of flagella, *H. pylori* motility, and TLR-mediated immune response could clarify gastric autoimmune and malignant conditions.

## Figures and Tables

**Figure 1 cancers-11-00648-f001:**
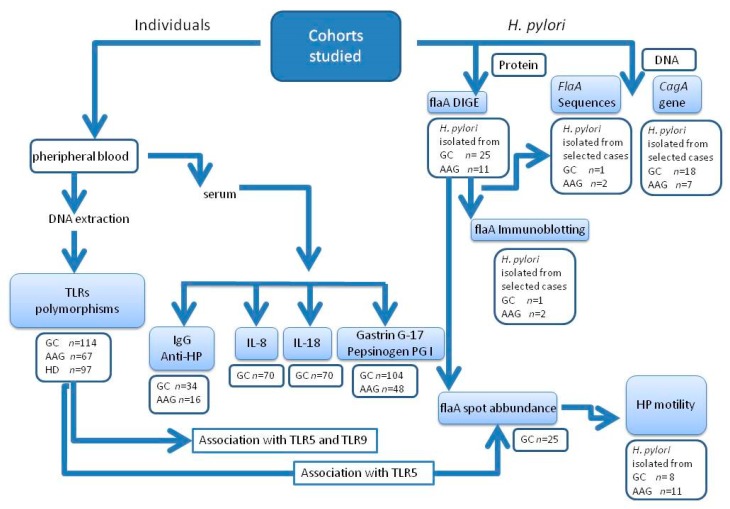
Flow diagram of the study design. 2 patients’ groups (GC and AAG), as well as healthy donors (HD) were analyzed for TLR polymorphisms (rs were listed in Table 2) and serological markers for *H. pylori* (HP) infection. *H. pylori*-related pro-inflammatory interleukin (IL)-8 and IL-18 cytokines were tested in GC and in association with the respective toll-like receptor (TLR)5 and TLR9 patient’s genotype. Gastrin G-17, PG-I and *H. pylori*-like markers for AAG and GC risk, were tested in association with the respective TLR5 and TLR9 patient’s genotype. The flagellin A of *H. pylori* strains isolated from AAG and GC patients were characterized by proteomics (DIGE (differential in gel analysis) and immunoblotting) and DNA sequencing. Spot abundances for flaA identified in each single gel by DIGE and bacteria motility were matched with the TLR5 polymorphism of the respective patient. The presence of virulent CagA gene in *H. pylori* strains isolated from patients with GC and AAG was tested by specific-polymerase chain reaction (PCR) assay. Number of individuals studied was reported under single assay.

**Figure 2 cancers-11-00648-f002:**
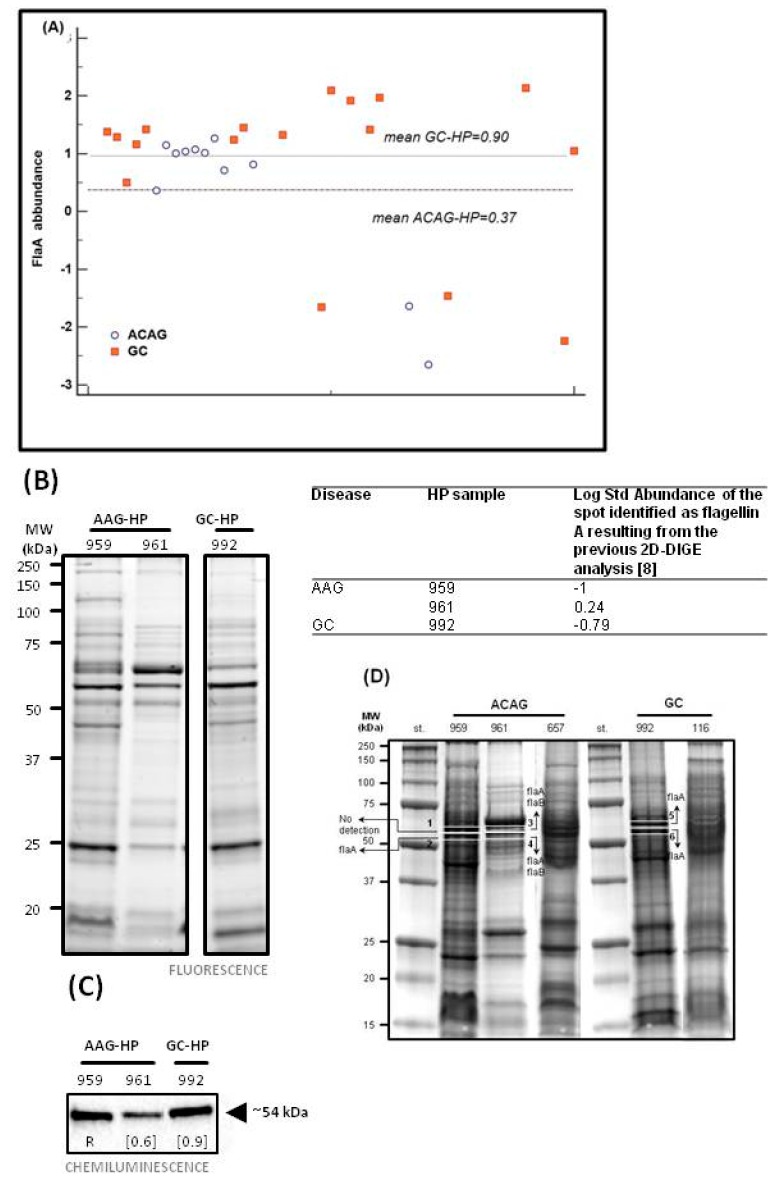
Proteomic characterization of flagellin A. (**A**) Abundance of flagellin A spot in a validation set of *H. pylori* isolates of 11 AAG patients compared to isolates of 18 GC. The mean value of flagellin A abundance was lower (0.37) in *H. pylori* isolates from ACAG than those previously obtained in a set of *H. pylori* isolates from GC patients (0.90) [6]. (**B**) One dimensional electrophoresis (1DE) strain-free gel acquired upon excitation with the Chemidoc system before its transfer to nitrocellulose membrane and (**C**) immunoblotting validation of flagellin expression in *H. pylori* isolated from either AAG (samples 959 and 961) and one GC (sample 992). Chemiluminescence signals of proteins cross-reacted with the anti-flagellin antibody were detected at around 54 kDa. Numbers refer to the relative quantity of the band calculated with the Image Lab TM software (R, reference band for which quantity is 1). Table report the details of protein abundance (Log Std Abundance) related to a two-dimensional electrophoresis (2D) spot previously identified as flagellin A in the 3 samples analysed [6]. (**D**) One dimensional electrophoresis (1DE) and identification of flagellins A (flaA) and B (flaB) in *H. pylori* isolated from either ACAG and GC. Two portions in the ~50–60 kDa area of the blue-stained 1DE gel were excised from each sample (AAG-HP: 959 and 961; GC-HP: 992) and submitted to analysis by mass spectrometry for protein identification. Arrows in the gel portions indicate the presence of flaA in all the samples and flaB in the only 961 one.

**Figure 3 cancers-11-00648-f003:**
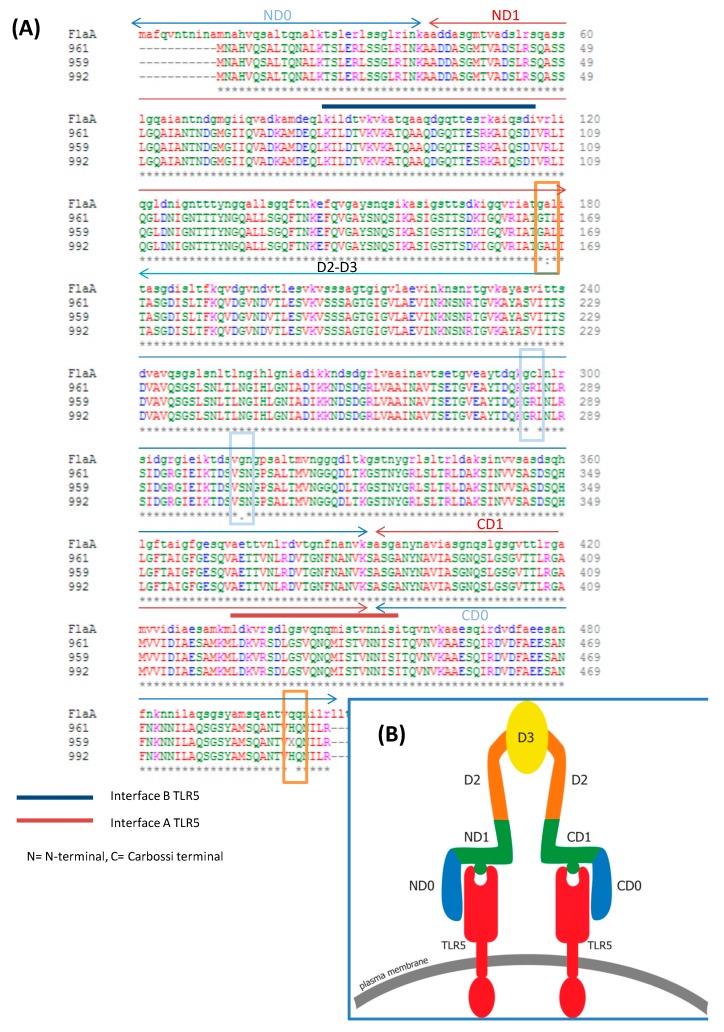
Multiple sequence alignment of flagellin A and schematic structure of flaA-TLR5 interaction. (**A**) The strains of *H. pylori* used as sources of DNA for FlaA sequencing were previously isolated from patients with GC and AAG. Two strains represent *H. pylori* isolated from the biopsy of AAG patient (sample 961 and 959), and one *H. pylori* strain isolated from the biopsy of a patient with GC (sample 992). Raw sequences of DNA samples were analyzed and compared to a reference sequence (FlaA) by using multiple BLAST CLUSTAL alignment analysis. (**B**) Schematic diagram of flagellin structure with respect to possible interactions with TLR5. D0 domain of flagellin is needed for the dimerization of TLR5 and it is recognized by both the C-terminal part of the TLR5 extracellular domain and the intracellular NOD-like receptor (NLRC4) of the inflammasome complex. Conserved residues in the D1 domain (indicated with blue and red lines in the figure) interface primarly with the groove TLR5 binding site.

**Figure 4 cancers-11-00648-f004:**
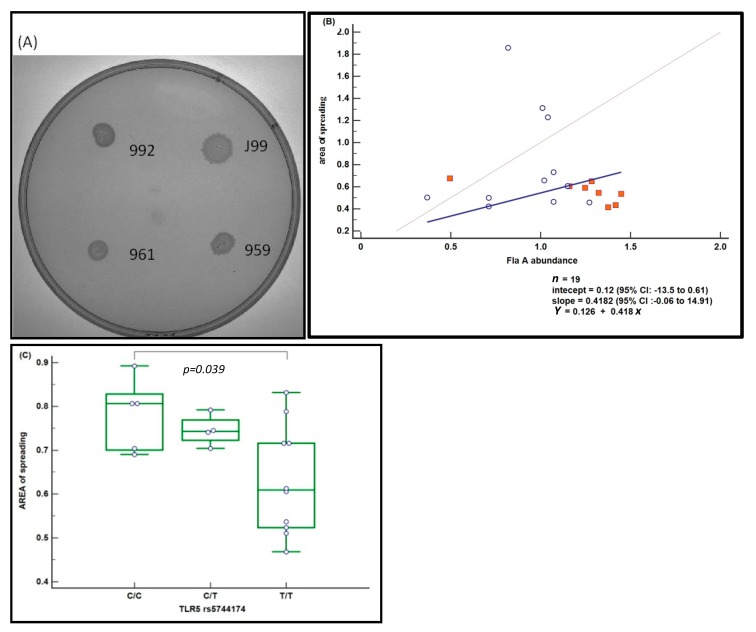
Motility of *H. pylori* (**A**) Bacteria were plated in soft agar and photographed after incubation for 7 days at the same multiplicity. Control: HPJ99, motile *H. pylori* strain J99 (ATCC700824) from patient with duodenal ulcer (**B**) Passing badlock regression plots for area of spreading surrounding each H. pylory colony (mm, mean of triplicate experiment) versus FlaA abundance. *H. pylori* were recovered from 11 AAG (hollow circle) and from 8 GC patients (filled squares). Dashed line represents the optimal regression line; solid line represents the best fit by linear regression including AAG and GC (*n* = 19) cases (*Y* = 0.12 + 0.42*x*). (**C**) Analysis of area of spreading versus TLR5 genotype corrected by FlaA abundance covariate.

**Figure 5 cancers-11-00648-f005:**
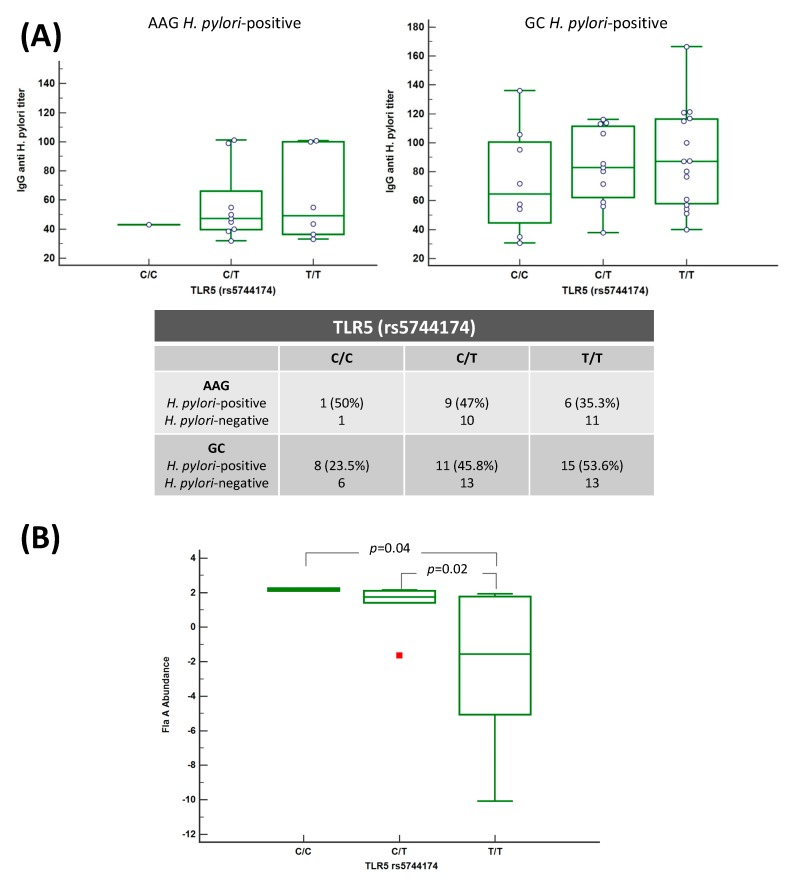
*H. pylori* IgG antibody titer and flaA abundance were associated with TLR5 genotype. (**A**) Figure illustrates the mean of *H. pylori* IgG titer stratified based on the presence of TLR5 rs5744174 genotype. Samples tested *n* = 100, 60 were GC and 40 AAG. IgG anti-*H. pylori* titer showed a slight tendency to increase in the genotype T/T, and in particular in the GC group of patients. (**B**) The mean of flagellin A abundance in *H. pylori* strain isolates is lower in the isolates of patients carrying the TLR5 rs5744174 genotype T/T.

**Figure 6 cancers-11-00648-f006:**
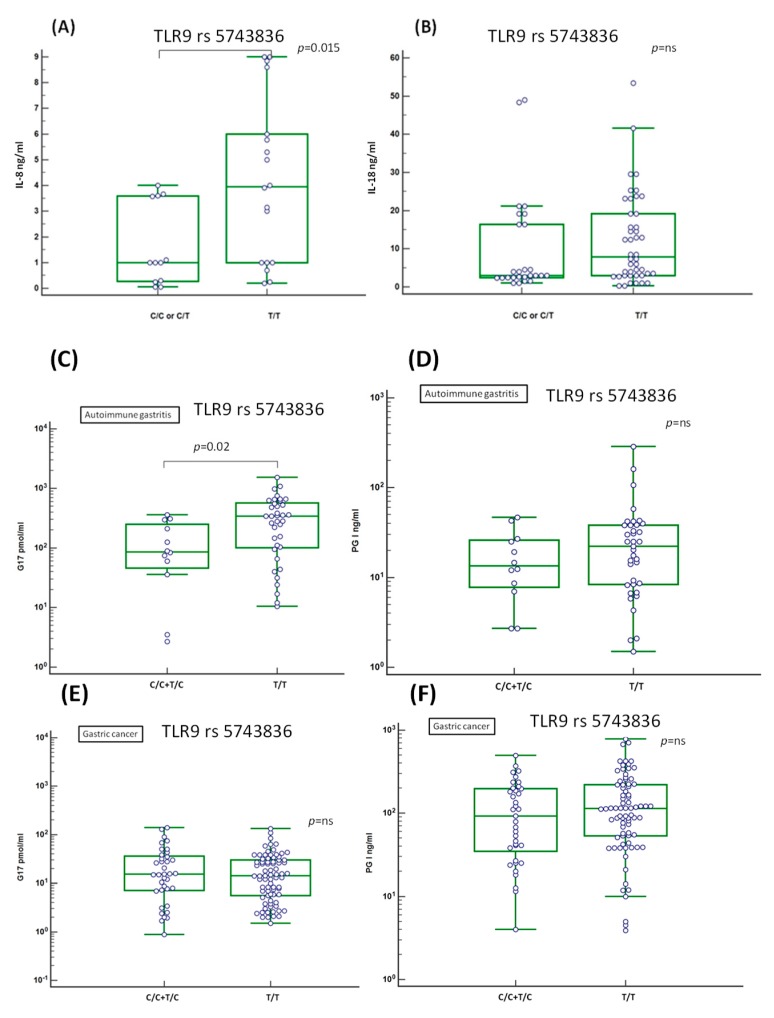
Cytokines, gastrin G17 and pepsinogen PGI association with TLR9 polymorphism. (**A**) IL-8 and (**B**) IL-18 cytokines titer were tested using a Luminex MagPlex based assay and result stratified based on the carriers TLR9 rs5743836 polymorphism. A significant increase in the mean IL-8 titer was found associated with the presence of the TLR9 T/T genotype (**A**). Samples tested were *n* = 70 GC, samples duplicated with a CV > 15% were excluded from the analysis and they are not reported in the figure. The mean of IL-18 levels between the 2 TLR9 genotypes (C/T and C/C vs. T/T) did not reach a statistical difference (**B**). TLR9 T/T genotype was found associated to highest mean level of gastrin G17 in the AAG group (**C**) but not in GC (**E**). TLR9 T/T genotype was found associated with a mean slightly higher PGI level in AAG (**D**) and in GC (**F**), although in both the groups the difference did not reach a significant difference. Compared to AAG, the mean of PG1 level is higher in GC.

**Figure 7 cancers-11-00648-f007:**
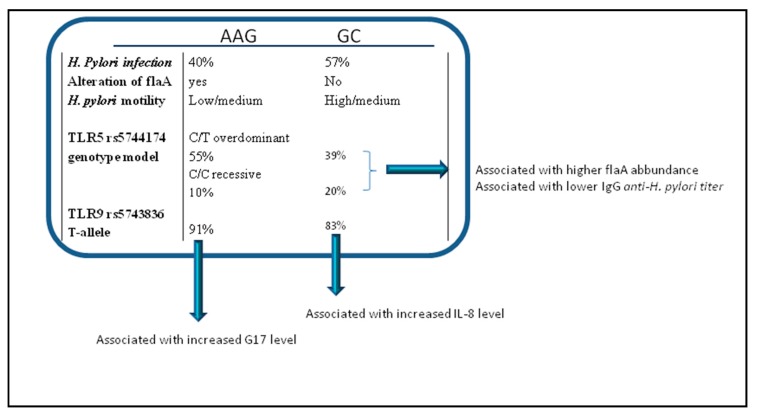
An abridged form of the different characteristics we observed in AAG and GC.

**Table 1 cancers-11-00648-t001:** Patient (GC *n* = 114, AAG *n* = 67) and healthy donors (HD *n* = 97) characteristics.

*Demographic features*	**Age at Inclusion**	**Mean ± SEM (Range)**		***p***
	GC	61.45 ± 1.04 (19–85)		GC vs. HD, *p* < 0.001
	AAG	54.59 ± 1.79 (31–70)		GC vs. AAG, *p* = 0.019
	HD	42.03 ± 1.66 (24–64)		AAG vs. HD, *p* < 0.001
	**Gender**	**Male**		
	GC	71 (62.3%)		GC vs. HD
	AAG	15 (22.4%)		GC vs. AAG, *p* < 0.001
	HD	55 (56.7%)		AAG vs. HD, *p* < 0.001
*H. pylori status*	***IgG Anti-H. Pylori Antibodies***	**positive**		
	GC	34 (56.7%)		GC vs. HD, *p* < 0.001
	AAG	16 (40.0%)		GC vs. AAG, *p* = 0.021
	HD	18 (18.6%)		AAG vs. HD
	Missed data	101		
*Pepsinogen*	**Pepsinogen I (ng/mL)**	**Mean** ± **SD**		
	GC	153.85 ± 137		GC vs. HD, *p* < 0.001
	AAG	37.29 ± 75		GC vs. AAG, *p* = 0.001
	HD	73.7 ± 89		AAG vs. HD
	Missed data	50		
	**Pepsinogen II (ng/mL)**	**Mean** ± **SD**		
	GC	22.58 ± 19		GC vs. HD, *p* < 0.001
	AAG	12.13 ± 6		GC vs. AAG, *p* = 0.007
	HD	7.2 ± 5		AAG vs. HD, *p* < 0.001
	Missed data	50		
	**PG I/PG II Ratio**	**Mean** ± **SD**		
	GC	6.96 ± 4.17		GC vs. HD
	AAG	2.43 ± 3.95		GC vs. AAG
	HD	9.57 ± 2.78		AAG vs. HD, *p* < 0.001
*Gastrin*	**Gastrin G17 (pg/mL)**	**Mean** ± **SD**		
	GC	20.47 ± 21		GC vs. HD, *p* = 0.003
	AAG	431.8 ± 504		GC vs. AAG, *p* < 0.001
	HD	7.7 ± 29		AAG vs. HD, *p* < 0.001
	Missed data	50		
*Tumor characteristics* (*n* = 60 GC)	**GC Lauren Classification**			
	Intestinal	30 (50.0%)		
	Diffuse	17 (28.3%)		
	Mixed	10 (16.7%)		
	unspecified	3 (5.0%)		
	**GC Anatomical Site**			
	Proximal	21 (35.0%)		
	Distal	38 (63.3%)		
	unspecified	1 (1.7%)		
	**Clinical Stage**			
	T	T1–T2 (26.7%)	T3–T4 (73.3%)	
	N	no (30.0%)	yes (70.0%)	
	M	no (82.5%)	yes (17.5%)	
	**Surgery**			
	Total gastrectomy	yes (40.7%)		
	Partial gastrectomy	yes (44.4%)		
	No surgery	yes (18.8%)		
*Autoimmune characteristics* (*n* = 47 AAG)	**Pernicious Anemia**	yes (29.4%)		
	***Familiarity for GC***	yes (12.7%)		

**Table 2 cancers-11-00648-t002:** Characteristics of selected Toll like receptors (TLR).

Gene	Chromosomal Position	Locus	SNP ID	Change Nucleotide AA		Reference
**TLR1**	4p14	exon 4	rs4833095	T > C	N248S	[8]
**TLR2**	4q32	5’UTR	-196 to -174del (rs111200466)	22-bp ins/del	---	[9]
		exon 3	rs3804099	T > C	N199N	[10]
**TLR4**	9q33.1	exon 3	rs4986790	A > G	D299G	[9,11]
		exon 3	rs4986791	C > T	T399I	[9,11]
**TLR5**	1q41	Exon 4	rs5744174	T > C	F616L	[12]
**TLR8**	Xp22.2	5´UTR	rs3764880	A > G	---	[13]
**TLR9**	3p21.3	5´UTR	rs5743836	T > C	---	[14]

**Table 3 cancers-11-00648-t003:** TLR primers.

Primers ID	Sequence (bp)	Size Products	References
TLR2-196, -174	FOR: CTCGGAGGCAGCGAGAAA	286/264	[60]
	REV: CTGGGCCGTGCAAAGAAG		
TLR4-EXON-3	FOR: TCTGCTCTAGAGGGCCTGTGC	632	(designed by the authors)
	REV: TCCTGGAAAGAATTGCCAGCCA		
TLR5-5744174	FOR: TGTCACTATAGCTGGGCCTC	104	(designed by the authors)
	REV: CCTCTTCATCACAACCTTCCG		

TLRs: Toll-like receptors; bp: base pair.

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
