# Peer review of "Polymorphism in Toll-Like Receptors and Helicobacter Pylori Motility in Autoimmune Atrophic Gastritis and Gastric Cancer"

_cancers, 2019, doi:10.3390/cancers11050648_

Reviewer 1 Report

This manuscript is tremendously difficult for readers to understand the relationship between TLR polymorphisms, Helicobacter pylori flagellin A gene mutation, autoimmune atrophic gastritis, and gastric cancer. The authors need to assemble better the contents of manuscript. In addition, the authors should drastically improve English language and carefully check out the miss typing.

The authors examined the motility of the three H. pylori strains 959, 961, and 992 isolated from the patients with either autoimmune atrophic gastritis or gastric cancer but did not mention the relationship between TLR polymorphisms and the H. pylori strains. What are the genotypes of TLR5 and TLR9 of those patients who were colonized by each H. pylori strain?

The bacterial name should be denoted with italic.

The number of bacteria should be denoted not with cells/ml but with colony-forming units (CFU)/ml.

Where are Tables 3 and 4?

Author Response

Ø  we reviewed the text, particularly the discussion, with the aim of facilitating the understanding of the study. The text has also been revised in the form and errors of English typing corrected. A certificate for this is attached.

Ø  A detailed table with data of motility, flaA and TLR genotype was added in figure 4.

Ø  Bacterial name is now denoted in italics in all the text

Ø  cells/ml has been corrected with colony-forming units (CFU)/ml.

Ø  table 3 and 4 were a refuse, we apologies for the mistake

Reviewer 2 Report

In the manuscript, “Polymorphism in Toll-like receptors and Helicobacter pylori motility in autoimmune atrophic gastritis and gastric cancer”, De Re and colleagues characterized TLR polymorphisms and features of bacterial flagellin A in samples from patients with autoimmune atrophic gastritis (AAG). They found that in a subset of patients alterations of flaA protein and two polymorphisms in TLR5 and TLR9 may favor the onset of AAG and gastric cancer.

The work is interesting even if the authors have to improve the following points:

·         When the authors described Table 1 they have to describe the difference in the expression of pepsinogen and gastrin that resulted statistically modified among the experimental groups

·        In Figure 2 the authors analyzed cases that have high or low flagellin A content compared with the mean value. What about a representative sample within the mean values?

·        Add the statistics in Figure 4B

·   Have the authors a correlation between flaA content and migratory potential? Please Increase the number of analyzed samples and verify  a possible correlation.

·        In Figure 6 A and B introduce the unit of measurement in the y axis

·     It would be useful for the readers to include the discussion of PMID: 14568977.

Author Response

We resubmit our manuscript entitled, “Polymorphism in Toll-like receptors and Helicobacter pylori motility in autoimmune atrophic gastritis and gastric cancer” (previous ID cancers-463096) for consideration as a Cancers research article in the special issue “Helicobacter Pylori associated cancer”.

We thank the reviewer for their appreciation of the study. With the aim of better illustrating the relationship between TLR5 polymorphism, flaA expression and H. pylori motility, in the present paper we have chosen to add some motility assay cases as suggested. We had to wait for the arrival of fresh reagents necessary for the assay, which is the reason for the delay in our response.

 Below please find more detailed responses  point by point.

Ø  We have detailed the difference in the expression of pepsinogens and gastrin in each single group (p column, Table 1)

Ø   We added to Figure 2 representative samples with mean value of flaA and added samples in figure 2A

Ø  We increased the number of samples for migratory assay in the Figure 4 and consequently we changed the choice of the statistical test (fig 4B).

Ø  We added the unit of measurement in the y axis in Figure 6 A and B

Ø  In the discussion we added the sentences, “Alterations in the immune response of patients with AAG may predispose to GC, and a link between AAG and GC has been reported in many studies. AAG is the outcome of a pathological CD4 T cell-mediated autoimmune response directed against the gastric H+/K+-ATPase and several epidemiological data point to a relation between immunogenetic background of the host and AAG development. A role for H. pylori as an antigen trigger for the genesis of AAG through a molecular mimicry between H. pylori antigens and gastric H/K-ATPase has been evoked, although definitive evidence to support this pathogen being needed for disease development still require more extensive investigation” and two references (ref 21: PMID: 14568977.; ref 22: PMID:21174235). 

Round  2

Reviewer 1 Report

Dear Authors,

The revised manuscript became to be more easy for readers to understand your work.

I think this revised version is suitable for publication in Cancers.

However, I still found the miss-typings in this version.

Please check again as for whether “H. pylori” is being written by Italic.

Please alter “Table 5” to “Table 3”.

Sincerely yours,

Reviewer 1

Author Response

Dear Editor and reviewer,

We resubmit our manuscript entitled, “Polymorphism in Toll-like receptors and Helicobacter pylori motility in autoimmune atrophic gastritis and gastric cancer” (previous ID cancers-463096) for consideration as a Cancers research article in the special issue “Helicobacter Pylori associated cancer”.

We thank the reviewers for their speed and accuracy of the revision.

We have corrected the H. pylori format in italics , Table 5 with Table 3 and revised the English mistakes.

Yours faithfully,

Dr. Valli De Re